# Genetic Identification and Traceability of Insect Meals

**DOI:** 10.3390/insects14070610

**Published:** 2023-07-05

**Authors:** Aristotelis Moulistanos, Nikoleta Karaiskou, Konstantinos Gkagkavouzis, Styliani Minoudi, Elena Drosopoulou, Chrysanthi Ioannidou, Nikolas Panteli, Stella Zografou, Damianos Karaouglanis, Dimitrios Kotouzas, Dimitrios Kontodimas, Efthimia Antonopoulou, Alexandros Triantafyllidis

**Affiliations:** 1Department of Genetics, Development and Molecular Biology, School of Biology, Aristotle University of Thessaloniki, 54124 Thessaloniki, Greece; amoulist@bio.auth.gr (A.M.); gagavou@bio.auth.gr (K.G.); sminoudi@bio.auth.gr (S.M.); edrosopo@bio.auth.gr (E.D.); xrusaioann@gmail.com (C.I.); karaouglanisdamianos@gmail.com (D.K.); atriant@bio.auth.gr (A.T.); 2Genomics and Epigenomics Translational Research (GENeTres), Center for Interdisciplinary Research and Innovation (CIRI-AUTH), Balkan Center, 57001 Thessaloniki, Greece; 3Department of Zoology, School of Biology, Aristotle University of Thessaloniki, 54124 Thessaloniki, Greece; nkpanteli@bio.auth.gr (N.P.); eantono@bio.auth.gr (E.A.); 4Department of Humanities, Social Sciences and Economics, School of Humanities, Social Sciences and Economics, International Hellenic University, 57001 Thessaloniki, Greece; s.zografou@ihu.gr; 5Laboratory of Agricultural Entomology, Benaki Phytopathological Institute, Kifissia, 14561 Athens, Greece; d.kotouzas@bpi.gr (D.K.); d.kontodimas@bpi.gr (D.K.)

**Keywords:** insects, aquafeed, COI gene, cloning, colony PCR

## Abstract

**Simple Summary:**

Insects that have been approved as suitable for consumption are an emerging feed source for aquaculture. Τo avoid the use of non-approved insects for the production of foods and feeds, analysis of the composition of products containing insects is deemed necessary. In this study, a genetic method of detecting the composition of insect meals was applied. Insect individuals were successfully identified, and the composition of nine insect meals was analyzed. As a result of this process, errors in labeling commercially available larvae were detected. No other insect species or animal species than reported were found in commercially available insect meals, while fungal species were identified in non-processed insect meal. Through this study, the use of DNA has been validated as a tool that increases the safety and quality of insect products.

**Abstract:**

Insects have been proposed as a rich alternative source of protein for the partial or total replacement of fishmeal in aquaculture. For maximum safety and effectiveness of insect meals, control of the quality composition of these products is considered mandatory. The aim of this study was the genetic analysis of the composition of commercially available insect meals at the species level. Commercially available *Hermetia illucens*, *Tenebrio molitor* and *Musca domestica* individuals, as well as nine insect meals produced from these species, were analyzed. The genetic identification of insects at the species level was based on a COI fragment, and analysis of the insect meals’ composition was performed with the processes of cloning and colony PCR. Genetic analysis indicated that the commercially available larvae morphologically identified as *Musca domestica* belonged to the species *Muscina stabulans*. In the commercially available insect meals, no other animal species was identified beyond the expected one. However, in the insect meal produced for research purposes, fungal growth was detected. The used methodology, herein, allows for the qualitative genetic identification of insect meals and could be included in the methods of traceability of products containing insects and other animal species.

## 1. Introduction

In recent years, insect meals have been increasingly studied and proposed as an alternative source of protein and nutrients for both humans [1] and animals [2,3]. Edible insects, such as *Tenebrio molitor*, *Locusta migratoria* and *Acheta domesticus*, in powder and other dried forms have recently been authorized by the European Commission as a novel food for human consumption [4]. This follows the approval of specific insects’ incorporation in feed for farmed animals. The European Commission, since 2017, has allowed the use of seven insect species, reared on specific substrates, as ingredients in aquafeeds (Regulation 2017/893/EC) [5] and poultry and pig feed (Regulation 2021/1372/EC) [6]. Despite the fact that insect consumption, also known as entomophagy, has been practiced for centuries as part of human culture, especially in Asia, Africa and Central and South America [1,7], Westernized societies are still hesitant and/or averse to insects as a nutrient source [8]. Insects, which also constitute part of several farmed species’ natural diet (e.g., pigs, cattle, poultry, fish) [9], are a dietary source of high nutritional value, with a sufficient content of energy, protein, fatty acids and vitamins [10,11]. Compared to conventional livestock production, insect rearing prevails in terms of environmental sustainability, due to lower emissions of greenhouse gases and ammonia and fewer resource requirements (e.g., water and land) [12,13]. Additionally, insects exhibit rapid reproduction rates and high fecundity and can feed on several agri-food and industrial wastes, effectively converting organic matter into nutrient biomass, while simultaneously adapting their amino acid and lipid profile [14,15,16,17]. The latter fits within the framework of circular economy concerning the minimization of waste and pollution, reuse of materials or wastes and regeneration of natural systems [18].

Dietary inclusion of insect meal in feed is currently being investigated in farmed species [3,19]. For example, several studies in fish species have demonstrated that fish meal can efficiently be substituted, especially at low rates of 25 to 30%, by insect meals without adverse effects on several parameters, including growth performance [2], nutrient digestibility [20], metabolic and antioxidant status [21], gut health and microbiota [22,23]. Regarding the two main species of the Mediterranean aquaculture, e.g., gilthead seabream (*Sparus aurata*) and European seabass (*Dicentrarchus labrax*), studies have focused on the inclusion of insect meals mainly from *Hermetia illucens* and *Tenebrio molitor* and, to a lesser extent, *Musca domestica* [2,23,24,25,26,27,28,29,30]. Although fewer studies have focused on terrestrial farmed species, insect meals, especially at low levels of inclusion, seem to exert no influence on growth performance and metabolism of pigs [3,31,32] and broiler chickens [19,33].

However, studies have shown that the use of different insect species on different farmed animals’ diet may have diverse effects depending on the species [30], which highlights the need to determine the species content in the insect meals prior to their utilization. In addition, the possible use of insect species, which have not been approved by the European Commission as suitable for feed, raises concerns and entails health risks, due to the lack of data on their safety [34]. Also, a key point about the quality of insect products is their microbiological load. In order to reduce pathogenic microorganisms and parasites, various processing methods with different effects on the final product are applied before the consumption of insects [35]. Therefore, both the insect species used for food and feed production and their subsequent processing seem to determine the safety and effectiveness of insect products, such as insect meals. This highlights the importance of accurately identifying their composition, as it appears to determine their safety and effectiveness. Historically, as is well known, the utilization of unauthorized animal products in feed has led to severe consequences, including the outburst of bovine spongiform encephalopathy (BSE) [36].

In order to ensure the safety of insect meals, effective and accurate methods should be employed for their analysis and their authentication, prior to the inclusion in animal feed. The application of genetic analysis methods to feeds has been a reliable approach for detecting species content. For example, real-time PCR has been successfully tested to identify *Tenebrio molitor* [34] and *Hermetia illucens* [37,38] in food and feed. Also, a multiplex-PCR for the amplification of 16S rRNA and COI genes has been used, for the successful identification of specific insect species, including *Tenebrio molitor*, in commercially derived insect samples (no insect meals) [39]. However, the identification of all possible species that may be contained in a product, such as insect meal, is not possible using the above methods where species-specific probes are used to target specific species. 

Non-targeted approaches based on cloning and next-generation sequencing (NGS) protocols allow for the analysis of the species composition, and sometimes, they can provide a quantitative aspect, even in processed products [40,41]. As regards insects, few cloning and NGS protocols have been developed to identify species in foods and feeds [42,43]. More specifically, a cloning protocol has been designed to identify insects in feeds, using a small region of the COI gene. From the implementation of this method, a large percentage of disagreements between expected and observed species was detected [42]. In addition, a DNA metabarcoding method based on a 16S rRNA fragment was carried out in commercially available products, and all the declared insect species were identified [43]. In the present study, we applied a non-targeted approach based on the COI gene, similar to the method developed by Garino et al., aiming for the traceability of commercially available insect meals that are used. The validation of this methodology in the present study highlights its ability for the qualitative analysis of insect meal composition at the species level, further ensuring the quality of food derived from insects.

## 2. Materials and Methods

### 2.1. Samples

Individual larvae of *Hermetia illucens*, *Musca domestica* and *Tenebrio molitor* and adult samples of Diptera from wild populations were used as positive controls for method development. More specifically, in the case of *Hermetia illucens* and *Tenebrio molitor*, larvae were collected from previous rearings of the Laboratory of Agricultural Entomology zoology at Benaki Phytopathological institute (BPI). In the case of *Musca domestica* larvae, samples were initially collected from the feed market. For the collection of new adult controls, plastic rearing trays (18 × 9 cm) were placed with control substrate (150 g wheat bran supplemented with fresh potato peels) and were left in the outdoor area of the laboratory at BPI for three days. The containers were subsequently immediately withdrawn to the lab rearing room. Each tray, in which dipteran eggs were found, was placed inside a (32 × 32 cm) rearing cage at optimal conditions (26 °C and 60–65% humidity). After the hatching of the first adults, samples were collected and sent for genetic identification. Evaluating the results of the genetic analysis, the desired species (*Musca domestica*) was isolated to be used for future production of biomass from similar larvae.

Eight commercially available insect meals (IM1–8) and one (IM 9) derived from rearing research of the Laboratory of Agricultural Entomology of BPI were analyzed in order to determine their species content (Table 1). According to labeling information of commercially available products, the IM 1–5 insect meals were prepared by *Hermetia illucens* and they were dried and defatted, the IM 6–7 meals included only *Musca domestica* and they were dried, and the IM 8, which was prepared using Tenebrio molitor, was processed with the freeze-dried method. The IM 9 insect meal also included *Tenebrio molitor* but was prepared with laboratory processing (air drying at 50 °C for 24 h). 

### 2.2. Sample Processing and DNA Extractions

DNA was extracted with a commercial extraction kit (QIAamp^®^ DNA Mini Kit, Qiagen, Hilden, Germany) both from controls and insect meals. The extracted DNA was preserved at −20 °C. Before the DNA extraction of insect meals, removal of fat and oils, independent of their processing method, was carried out. This step was added because the lipid concentration of insects [44] could interfere in DNA extraction process [45] and inhibit PCR [46]. The fat and oil extraction process involved resuspending 10 mg of sample in methanol–chloroform–water (2:1:0.8) for one day and then washing in distilled water and PBS 1× buffer to eliminate the remains of the solution previously used.

During method development, three pairs of primers were tested: two pairs for regions of the COI gene with different lengths (130 bp and 650 bp) and one for an 80–125 bp fragment of the 16S rRNA gene [47,48,49], in order to select the most suitable region for the identification of insects at species level. The total volume of the polymerase chain reaction (PCR) was 30 μL, in which 100 ng of genomic DNA was amplified, using 0.05 units of Qiagen Taq polymerase, 2 mM dNTPs, 0.3 μL of each primer (100 μΜ), 2.5 mM MgCl_2_ (Qiagen, Hilden, Germany) and 3 μL of 10X Reaction Buffer (Qiagen, Hilden, Germany). PCR amplification conditions of each region are described in Table 2. Amplification products were assessed via electrophoresis in 1.5% (*m*/*v*) agarose gels.

The successfully amplified products of insect meals were purified with PureLink PCR Purification kit (Invitrogen Life Technologies, Carlsbad, CA, USA) and cloned with Qiagen PCR Cloning kit (Qiagen, Hilden, Germany) following manufacturer instructions. Thus, 30 to 50 clones from each insect meal were amplified with colony PCR, using the universal plasmid primers SP6 (5′-ATTTAGGTGACACTATAG-3′) and T7 (5′-TAATACGACTCACTATAGGG-3′). The amplification reaction was performed in a total volume of 30 µL using the same concentrations as above. PCR conditions were as follows: lysis of bacterial cells and release of plasmid DNA from the cells at 94 °C for 25 min, denaturation at 94 °C for 5 min, 40 cycles of denaturation at 94 °C for 30 s, annealing at 51 °C for 4 min and extension at 72 °C for 1 min and a final extension at 72 °C for 10 min. 

All successfully amplified products, from control samples and colony PCR, were sent to the company Genewiz (Takeley, UK) for purification and sequencing. Sequences obtained were checked using Geneious software version 10.2.6 and were analyzed employing the program BLAST within NCBI database: https://blast.ncbi.nlm.nih.gov/Blast.cgi (accessed on 4 July 2023). Sequence similarity > 98% was considered reliable [50].

## 3. Results

### 3.1. DNA Fragment Selection for Genetic Identification

Of the studied regions in the larvae and adult samples, the 650 bp fragment of the COI gene amplified successfully in all species. PCR products of the smaller regions of the COI and 16 rDNA genes were of low quality. In the case of *Hermetia illucens* samples, no amplification for 16S rRNA was achieved. Therefore, the 650 bp COI fragment was chosen as the most suitable region for analyzing the species composition of insect meals.

Genetic data verified morphological identification for all commercially available larvae analyzed from *Hermetia illucens* and *Tenebrio molitor* (four larvae from each species). However, all six larvae morphologically characterized as *Musca domestica* were genetically identified as *Muscina stabulans,* with >99% similarity (Table 3). Ιn addition, the genetic identification was carried out for adult samples from wild populations of Diptera, which were going to be used for rearing and production of insect meals by the BPI. In these cases, one individual belonged to the species *Musca domestica* (Table 3). 

### 3.2. Analysis of the Insect Meals

In total, more than 40 sequences were identified for each of the IM 1–7 insect meals, while for IM 8 and 9, the number of analyzed sequences was 34 and 20, respectively. Detailed results of sequencing clones per insect meal are shown in Table 4.

In all studied commercially available insect meals, no other species was identified than the one reported on the label. In IM 9, which was derived from research rearing, 19 of the 34 clones sequenced (55.9%) were identified as *Tenebrio molitor*, and the rest were found to belong to fungal species. More specifically, two of them were identified as *Candida* sp., with a low identity of 80%, and the rest as *Millerozyma farinose*, with an identity of 99.47%. 

## 4. Discussion

### 4.1. Suitability of the Selected DNA Region

The COI mitochondrial gene is a key molecular marker for animal species identification and is also widely used in a variety of insect species identification projects [51,52,53,54,55,56,57], resulting in many deposited sequences in databases. More specifically, the fragment of the COI gene, which was selected in the present study, has been proven to enable the identification of many insect species, including *Hermetia illucens*, *Tenebrio molitor* and *Musca domestica* [47,57].

Its species discriminating potential has been further confirmed in this present study by the successful distinction between *Musca domestica* and *Muscina stabulans* samples, revealing a conflict between morphological and genetic identification. This finding detected the existence of commercially available species that have not been approved from 2017/893 to be placed on the feed market (*Muscina stabulans* as *Musca domestica*). In cases in which morphological identification is known to present difficulties [53], such as within the Muscidae family, the genetic identification of insects appears to be necessary, particularly in the field of feeds and foods, in order to avoid unknown effects from the consumption of misidentified species that have not yet been investigated or approved.

Nevertheless, a possible disadvantage of the amplified region, used in the present study, is its long size (650 bp), as, in insect meals where the product has undergone some processing and DNA may be fragmented, it is recommended to target shorter regions (<200 bp) [58]. However, despite the various types of processing of IM 1–8 insect meals, successful detection of the declared species was carried out in all insect meals; thus, the processing method applied for the production of insect meals (drying or defatting) does not seem to interfere with the amplification of the targeted COI fragment.

### 4.2. Reliability of the PCR-Cloning Methodology

As mentioned above, the PCR-cloning methodology has been used successfully to analyze the composition of both fish and insect meals [40,42]. This approach allows for the species identification of individuals used for the preparation of insect meals; therefore, it allows for the control of their composition. In the present study, the cloning of insect meal samples showed that IM 1–8 did not have any other insect or animal species in general, apart from the one that was declared. Agreement between the reported and identified composition was also observed by Hillinger et al. in food products from insects [43]. 

Despite the somewhat smaller number of sequences cloned and analyzed in IM 9, the contamination with fungi species (*Millerozyma farinosa*, *Candida* sp.) was successfully detected. Although there is a specific DNA barcode locus for fungi identification, such as ITS region, the COI gene also allows, to some extent, for the detection of fungi species [59,60]. For instance, in IM 9, identification at the species level could not be precisely supported in the case of *Candida* sp., because of the low similarity (80%) with deposited sequences in the NCBI database. However, in the case of *Millerozyma farinosa,* 13/34 sequences were identified to the species level with high similarity (99.47%), so the methodology is partly capable of identifying fungi to the species level. Both *Millerozyma farinosa* and many *Candida* species are symbiotics of insects [61], as, in insects, there are suitable conditions (moisture and nutrients) for the survival and growth of microorganisms [62]. In this meal, the larvae were prepared with laboratory-scale processing, thus favoring, to some extent, the development of fungi. This finding highlights the need for proper maintenance and processing of insect products and its importance for their quality and safety. The lack or use of the incorrect processing method can also favor the increase in the concentration of bacteria, such as *Staphylococcus aureus* and *Salmonella* spp. [35], which are not easy to detect using the present method. Therefore, the development of a genetic method to accurately identify microorganisms in insect products is considered essential to control their quality. 

### 4.3. Νon-Targeted Genetic Analyses of Insect Meal Composition

In a relevant study by Garino et al., animal feeds from insects, which were purchased either online or from local reptile feed shops, were analyzed using the method of cloning and COI gene [42]. The results of the analysis showed that there is a large percentage of unexpected species in the composition of the studied products [42], but this finding was not detected in the insect meals of the present study. This difference is largely due to the fact that, in the present study, the commercial samples were from companies, and their composition was declared, while in the work of Garino et al., some of the tested samples did not have a certified origin, and the indication of the species was given only by the seller [42]. This does not necessarily suggest that those were cases of adulteration but highlights the difficulties in the morphological identification of insect species.

Another approach for the composition analysis of insect meals is the metabarcoding with next-generation-sequencing-based molecular method (NGS). Protocols have been designed based on DNA metabarcoding analysis to identify insects in products, using either a small region of the COI gene [42] or a 16S rRNA fragment [43]. The different methods, which can be applied to insect products, allow for the discovery of new study targets to produce high-quality edible insects [63]. The development of analytical methodologies to control the quality and quantity composition of insect products is a necessary step for the verification of their safety and authenticity, with the ultimate goal of avoiding unintended side effects, such as allergies or diseases, for the protection of consumers’ health [43]. 

## 5. Conclusions

Insects appear to be a rising source of protein for aquaculture. The effects of feeding specific insect species to various fish species have been a key research question in recent years. The number of insect species, which has been studied and approved as suitable aquafeed, is limited. Therefore, control of the exclusive use of these species to produce insect meals is necessary. The method used in this study showed that PCR cloning of the COI gene enables the identification and detection of animal species, which are contained in insect meals, as well as even fungi that may have grown within them. The results highlight two peculiarities associated with the production of insect meals that require attention: (i) the assurance of accurate insect identification before rearing and (ii) the need for proper maintenance and processing of their products to avoid contamination. 

## Figures and Tables

**Table 1 insects-14-00610-t001:** Details about the studied insect meals.

Insect Meal (IM)	Declared Composition	Type of Processing	Origin
IM 1	*Hermetia illucens*	Dried/Defatted	Baruth/Mark, Germany
IM 2	*Hermetia illucens*	Dried/Defatted	Baruth/Mark, Germany
IM 3	*Hermetia illucens*	Dried/Defatted	Baruth/Mark, Germany
IM 4	*Hermetia illucens*	Dried/Defatted	Baruth/Mark, Germany
IM 5	*Hermetia illucens*	Dried/Defatted	Nesle, France
IM 6	*Musca domestica*	Dried/Non-Defatted	Russia
IM 7	*Musca domestica*	Dried/Non-Defatted	Russia
IM 8	*Tenebrio molitor*	Freeze-dried/Non-Defatted	Ermelo, The Netherlands
IM 9	*Tenebrio molitor*	Air-dried/Non-Defatted	Benaki Phytopathological Ιnstitute

**Table 2 insects-14-00610-t002:** Fragment size and PCR conditions for all tested primer pairs.

Primers	Gene	Fragment Size	PCR Conditions	Reference
16S-HF	16S rDNA	80–125 bp	95 °C–5 min/(95 °C–20 s, 58 °C–20 s, 72 °C–30 s) × 35 cycles/ 72 °C–20 min/20 °C–1 min	Horreo et al., 2013 [48]
16S-HR				
Uni-MinibarR1	COI	130 bp	95 °C–2 min/(95 °C–1 min, 46 °C–1 min, 72 °C–30 s) × 5 cycles/ (95 °C–1 min, 53 °C–1 min, 72 °C–30 s) × 35 cycles/72 °C–5 min	Meusnier et al., 2008 [49]
Uni-MinibarF1				
LepF1	COI	648 bp	94 °C–1 min/(94 °C–1 min, 45 °C–1 min 30 s, 72 °C–1 min 15 s) × 6 cycles/(94 °C–1 min, 51 °C–1 min 30 s, 72 °C–1 min 15 s) × 36 cycles/72 °C–5 min	Hebert et al., 2004 [47]
LepR1				

**Table 3 insects-14-00610-t003:** Origin, morphological and genetic identification of studied larvae and adult samples.

Number of Samples/Stage of Life	Morphological Identification	Genetic Identification	Sequence Similarity with Deposited in NCBI Database
4/larvae	*Hermetia illucens*	*Hermetia illucens*	100%
4/larvae	*Tenebrio molitor*	*Tenebrio molitor*	100%
6/larvae	*Musca domestica*	*Μuscina stabulans*	99.77%
1/adult	*-*	*Musca domestica*	100%
1/adult	*-*	*Sarcophaga africa*	100%

**Table 4 insects-14-00610-t004:** Results of genetic analysis at species level of insect meals. The declared composition and the number of identified/amplified clones are also presented.

Insect Meal (IM)	Declared Composition	Identified/ Amplified Clones	Result of Genetic Analysis
IM 1	*Hermetia illucens*	45/48	*Hermetia illucens*
IM 2	*Hermetia illucens*	49/53	*Hermetia illucens*
IM 3	*Hermetia illucens*	41/46	*Hermetia illucens*
IM 4	*Hermetia illucens*	42/48	*Hermetia illucens*
IM 5	*Hermetia illucens*	46/52	*Hermetia illucens*
IM 6	*Musca domestica*	47/51	*Musca domestica*
IM 7	*Musca domestica*	43/46	*Musca domestica*
IM 8	*Tenebrio molitor*	20/35	*Tenebrio molitor*
IM 9	*Tenebrio molitor*	34/39	*Tenebrio molitor*
			Fungi species

## Data Availability

The data presented in this study are available on request from the corresponding author.

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
