# Peer review of "Genetic Identification and Traceability of Insect Meals"

_insects, 2023, doi:10.3390/insects14070610_

Round 1

Reviewer 1 Report

General comments:

Overall, the work presented in the manuscript is heavily based on work outlined and methods developed in Reference 31. Authors fail to mention the similarity to this work, or contrast the novelty of the methods they are presenting in the manuscript relative to previous work. This should be clarified in the text. Line 214 makes some comparison to this reference in discussion but this should be elaborated in introduction or methodology as it is the basis for the work described in this manuscript.

Simple summary is not a simplified reiteration of the abstract and fails to communicate a simplified version of the same information.

·        Line 19 For example, the goal and purpose of insect production for animal feed or human food mentioned in the simple summary versus Line 32 focus on aquaculture as fishmeal replacement. Additionally, I suggest further

discussion of insect meal use outside of aquaculture alone.

The authors discuss the importance and impact of processing, stating this as one of their major conclusions (L240) but do not provide any details on processing method nor the basis for their conclusion of it’s importance. Much more detail is needed to evaluate their conclusion.

Specific comments:

L 30 This study has further validated the use of DNA barcoding as tool, rather than the emergence of a new tool.

L 56, Change “insects” to “insect”

L 83, Change “16S rDNA” to “16S rRNA” as this is the gene, correction need to be made throughout the manuscript

L 91, Work referenced in this manuscript is highly based on Reference 31, and this seems to be a validation or repetition of that work, rather than a novel experiment

L 99, “Going to reared” – sentence should be rewritten for clarity

L 100, more detail should be provided on the insect meals, including processing method, particularly since the authors suggest that they were processed differently (L191). 

L 105, Table 1 Provide reason for failure of identification of last two entries. Why were they not identified morphologically?

L 108, Change “DNA Analysis” to “Sample Processing and DNA Extractions” or alternative subtitle that better represents methodology in this section

L 111-114 Where was the method for lipid extraction derived? Is there a reference for this ?

L 114, Authors fail to mention at what point of the sample processing the DNA was extracted, and goes from fat/oils extraction to preservation of extracted DNA. Methods should explicitly include sample processing steps.

L 126, Table 2 Include “Fragment Size” unit, e.g. bp

L 144, Change “Methods Development” to alternative subtitle that better represent results discussed in this section

L 158, Incorrect formatting for subtitle

L 166, The identification of fungi at the species level is effectively done using ITS gene, as this is the standard barcode for fungi. The use of COI gene for DNA barcoding is proposed and reserved for animal species identification. Therefore, I would discount the identification of fungal species using the primers used in this manuscript, unless otherwise validated using the ITS gene.

L 186, Provide citation supporting this claim based on differentiations of previously compared species in the same genus.

L 193 No declaration on fungal growth identification should be made unless validated using ITS gene, which has not been done in study described in this manuscript.

The manuscript needs further review and editing to rectify issues with sentence structure and grammatical errors that make the comprehension of the manuscript difficult. Common errors throughout the document include run-on sentences, misuse of coordinating conjunctions and commas, to name a few examples.

Author Response

We believe that the revisions resulted in an improved revised manuscript, which you will find uploaded marked with track changes. The number of lines correspond to the Manuscript with track changes. The manuscript has been revised to address the reviewer comments.

Reviewer 2 Report

dear authors,

the article deals with a very interesting and current topic. It is well structured and presents a good readability. I suggest you some improvement reported belowe

In the introduction (lines 92-93) you stated that the aims of this work is to perform a qualitative analysis of insect based feed to grant the quality food derived from insect and fish, but, really, all reported  results are related to feed quality and not to the qualitative features of foods obtained by insect meals.

I suggest to better define the aims of the study

I would suggest to broaden the discussion of the results with some considerations about the positive effects of the availability of analitic methodologies ensuring the quality of insect based aquafeeds, about the potencial recipients of these innovative tools, and, in general, about the positive contribution to the feed and food safety system

Introduction: line 88-89 please reformulate: the verb seems missing

Conclusions: lines 232-233 This sentence is not clear to me, please reformulate.

Author Response

We believe that the revisions resulted in an improved revised manuscript, which you will find uploaded marked with track changes. The number of lines correspond to the manuscript with track changes. The manuscript has been revised to address the reviewer comments.

Point 1: In the introduction (lines 92-93) you stated that the aims of this work is to perform a qualitative analysis of insect based feed to grant the quality food derived from insect and fish, but, really, all reported  results are related to feed quality and not to the qualitative features of foods obtained by insect meals.

Response 1: Done (Line 122-124). We understand that the aim was not stated clearly. The sentence was therefore rephrased “The validation of this methodology in the present study highlights its ability for qualitative analysis of insect meals composition at species level, further ensuring the quality of food derived from insects.”.

Point 2: I suggest to better define the aims of the study

Response 2: Done. The whole paragraph (Lines 116-124) was rephrased.

Point 3: I would suggest to broaden the discussion of the results with some considerations about the positive effects of the availability of analitic methodologies ensuring the quality of insect based aquafeeds, about the potencial recipients of these innovative tools, and, in general, about the positive contribution to the feed and food safety system.

Response 3: Sentences were added (Lines 319-327).

Point 4: Introduction: line 88-89 please reformulate: the verb seems missing.

Response 4: Done. The whole paragraph (Lines 116-124) was changed.

Point 5: Conclusions: lines 232-233 This sentence is not clear to me, please reformulate.

Response 5: Done. The whole paragraph (Lines 318-327) was rephrased.